# Liver Transplantation from Elderly Donors (≥85 Years Old)

**DOI:** 10.3390/cancers16101803

**Published:** 2024-05-08

**Authors:** Pierluigi Romano, Luis Cano, Daniel Pietrasz, Nassiba Beghdadi, Marc-Antoine Allard, Chady Salloum, Frédérique Blandin, Oriana Ciacio, Gabriella Pittau, René Adam, Daniel Azoulay, Antonio Sa Cunha, Eric Vibert, Luciano De Carlis, Alessandro Vitale, Umberto Cillo, Daniel Cherqui, Nicolas Golse

**Affiliations:** 1Department of Hepato-Biliary and Pancreatic Surgery, and Liver Transplantation, Paul Brousse Hospital, AP-HP, 94800 Villejuif, France; romano.pierluigi5@gmail.com (P.R.); nassiba.beghdadi@hotmail.fr (N.B.); oriana.ciacio@aphp.fr (O.C.); gabriella.pittau@aphp.fr (G.P.); rene.adam@aphp.fr (R.A.);; 2Department of Surgical, Oncological and Gastroenterological Sciences (DiSCOG), Second General Surgical Unit, University of Padova, 35122 Padua, Italycillo@unipd.it (U.C.); 3Department of General and Transplant Surgery, Grande Ospedale Metropolitano Niguarda, School of Medicine and Surgery, University of Milan-Bicocca, 20126 Milan, Italy; luciano.decarlis@ospedaleniguarda.it; 4INRAE, CHU Pontchaillou, UMR 1241 NUMECAN, Université de Rennes, 35033 Rennes, France; 5INSERM, Physiopathogénèse et Traitement des Maladies du Foie, UMR-S 1193, Université Paris-Saclay, 94805 Villejuif, France

**Keywords:** liver transplantation, donor age, propensity score matching, outcomes, elderly donors

## Abstract

**Simple Summary:**

This study examines the viability of liver grafts from donors aged 85 years and older in liver transplantation (LT) compared to those from younger donors under 40 years old. The research, conducted on data from 2005 to 2023, evaluates post-LT outcomes using propensity score matching. Despite lower 5-year survival rates in the elderly group before matching, the proposed nomogram provides a more acceptable 10-year post-LT survival using grafts from older donors. Notably, the study emphasizes the importance of proper matching, particularly for recipients with hepatocellular carcinoma (HCC), in achieving satisfactory long-term results amid organ scarcity.

**Abstract:**

Background: Despite the ongoing trend of increasing donor ages in liver transplantation (LT) setting, a notable gap persists in the availability of comprehensive guidelines for the utilization of organs from elderly donors. This study aimed to evaluate the viability of livers grafts from donors aged ≥85 years and report the post-LT outcomes compared with those from “ideal” donors under 40 years old. Methods: Conducted retrospectively at a single center from 2005 to 2023, this study compared outcomes of LTs from donors aged ≥85 y/o and ≤40 y/o, with the propensity score matching to the recipient’s gender, age, BMI, MELD score, redo-LT, LT indication, and cause of donor death. Results: A total of 76 patients received grafts from donors ≥85 y/o and were compared to 349 liver grafts from donors ≤40 y/o. Prior to PSM, the 5-year overall survival was 63% for the elderly group and 77% for the young group (*p* = 0.002). After PSM, the 5-year overall survival was 63% and 73% (*p* = 0.1). A nomogram, developed at the time of graft acceptance and including HCC features, predicted 10-year survival after LT using a graft from a donor aged ≥85. Conclusions: In the context of organ scarcity, elderly donors emerge as a partial solution. Nonetheless, without proper selection, LT using very elderly donors yields inferior long-term outcomes compared to transplantation from very young donors ≤40 y/o. The resulting nomogram based on pre-transplant criteria allows for the optimization of elderly donor/recipient matching to achieve satisfactory long-term results, in addition to traditional matching methods.

## 1. Introduction

Liver transplant (LT) is the best treatment for end-stage liver diseases and in patients diagnosed with hepatocellular carcinoma (HCC) alongside cirrhosis [1]. LT stands as a crucial life-saving procedure globally, with a growing demand for organs amidst a scarcity of donors. There is currently a shortage of organs available for patients on the waiting lists, which means the mortality rates of patients on the waiting list are increasing, as well as dropout rates [2,3]. As an example, in 2020, in France, over 1841 new patients were listed, revealing a stark contrast to the 1128 LTs performed, with 38% originated from donors aged over 65 (only 6% in 2000), resulting in about 800 patients still awaiting LTs at the end of the year. Moreover, due to the development of transplant oncology and possible new indications for LT (HCC after downstaging, intra-hepatic cholangiocarcinoma, and unresectable colorectal liver metastases), the gap between grafts and needs may soon worsen [3].

The organ shortage has prompted considerations of expanding the donor age criterion, despite regional variations in acceptable donor ages [4,5,6,7]. European studies have supported evaluating liver grafts without age restrictions, leading to an increase in older grafts from 0.1% in 2000 to 3.4% in 2013. Reviews of the literature demonstrated comparable short- and medium-term graft and patient survivals in octogenarian LT, emphasizing the value of older donors despite the increased susceptibility to biliary complications [8,9,10,11,12]. Some studies advocated for the careful selection of octogenarian grafts to broaden the pool of safe donors, which was supported by meticulous evaluation, selective matching, and skilled perioperative care [13,14,15,16,17,18,19].

Conversely, the donor age significantly impacts post-LT prognostic scores like the Donor Risk Index (DRI), the BAR score, and the Donor Model of End-Stage Liver Disease (D-MELD), resulting in the decreased utilization of grafts from older donor in the U.S. [4].

The appropriateness of utilizing older grafts, the required matching, as well as the “old graft” definition remain challenging. Recent data, however, suggested the potential for a more liberal use of older liver grafts to address organ shortage issues, potentially influencing graft acceptance policies, Organ Procurement Organizations, and transplant logistics [12,15,16,20]. The aim of the present study was to assess the feasibility of utilizing grafts from donors aged over 85 years old and compare their outcomes with “ideal” grafts from donors under 40 years old to better highlight the effects of age, if any. The choice of the upper age limit was motivated by the lack of specific analyses in the literature on donors over 85 years old and the need to evaluate our policy on the acceptability of very old grafts. This study sought to investigate the practicality and viability of employing grafts from elderly donors, a population traditionally considered of marginal interest for organ donation. Additionally, our objective was to establish a practical “nomogram” that could facilitate the careful matching of very old grafts with corresponding suitable recipients in addition to the actual selection methods.

## 2. Patients and Methods

### 2.1. Study Design and Study Population

A retrospective analysis was carried out on all the LTs performed using liver grafts from brain-dead donors aged ≥85 (study group) or ≤40 years old (control group) between September 2005 and January 2023 [21]. Grafts from donors after circulatory death (DCDs), living donors, split/reduced grafts, and auxiliary transplantations were not included. The results of the study group (n = 76) were compared with those of the control group (n = 349). Propensity score matching (PSM) was performed afterwards.

### 2.2. Donor Evaluation

Elderly donors were preliminarily evaluated based on their past medical history, including substance abuse, smoking, diabetes/cardiac/renal/pulmonary diseases, presence of hypertension and oncological history, blood tests, and CT scan features (imaging available for all donors), and were eventually declined if one of the following was present (case by case decision tailored to the recipient’s status): major cardiovascular history, prolonged cardiac arrest, increasing transaminase/bilirubin/gamma-glutamyl transferase (GGT) values, major steatosis suspected at unenhanced CT scan, obesity, history of major alcohol abuse, predictable long cold ischemia time, etc. Once accepted, a liver graft biopsy at procurement was systematically performed (for a postponed analysis), and a frozen section was analyzed in case of graft quality doubt and analyzed extemporaneously if logistical conditions permitted. Every effort was then made to synchronize the teams and keep the duration of cold ischemia to a minimum.

### 2.3. Graft Allocation Policy

In France, grafts are allocated to a specific recipient by the National Biomedicine Agency according to the severity of the liver disease (MELD score) and points awarded in the presence of HCC within the AFP score [17]. After graft refusal by several teams, it is then possible to accept an “hors tour” graft, which is allocated to a team and no longer to a specific patient. Our policy for the use of elderly grafts (≥85 years) is fundamentally aimed at prioritizing patients with HCC not or insufficiently controlled (high risk of dropout) or any recipient with poor access to a graft (no prioritization).

### 2.4. Data Collection and Definitions

In the data analysis, the following donor-related variables were collected: age, sex, cause of death, stay in the intensive care unit (ICU), the need for vasoactive drugs before and during organ procurement, and test values of aspartate aminotransferase (AST), alanine aminotransferase (ALT), total bilirubin, GGT, prothrombin activity, and plasma sodium. Surgical variables in the recipient included the following: cold ischemia times of the graft, type and amount of transfusion, duration of the LT, age, sex, body mass index, indication for LT, CHILD, and MELD scores, time on the waiting list, and standard postoperative blood analysis.

The MELD score was assessed for all recipients and was part of the French liver allocation system. The donor MELD (D-MELD) score, shown to be associated with short- and long-term outcomes (with poorer outcomes for D-MELD scores over 1600), was also calculated [18].

Primary non-function (PNF) was defined as the need for urgent re-transplantation within 10 days when a graft did not demonstrate any evidence of initial function following transplantation, after the exclusion of other causes like hepatic artery thrombosis or acute cellular rejection. These patients developed high transaminases, a low prothrombin time, high bilirubin, and high lactate within 24 h after liver transplantation [22]. Early allograft dysfunction (EAD) was defined by the presence of at least one of the following criteria, as proposed by Olthoff et al. [23]: bilirubinaemia levels of 170 μmol/L (10 mg/dL) at day 7, international normalized ratio (INR) of 1.6 or greater at day 7, and transaminases ≥ 2000 UI/L within the first 7 days. Acute rejection was classified using the Banff grades [24]. Total biliary complications included early complications (fistula and early stenosis) and late complications (anastomotic and non-anastomotic stenosis). Vascular complications were defined as all post-transplant abnormalities in the hepatic artery, portal vein, or vena cava requiring therapeutic procedures. The primary outcome was one, three, and five-year graft and patient survivals following LT. We also focused on any adverse outcomes.

The present cohort study has been written following the STROBE guidelines [25].

### 2.5. Statistical Analysis

The qualitative data are expressed as frequencies (%), and the quantitative data as the median [interquartile range Q1, Q3]. The statistical significance for group comparisons was set at *p* < 0.05. The survival curves were analyzed using the log-rank test and the Gehan–Breslow–Wilcoxon method (giving more weight to events at early time points) before and after PSM.

Univariate analysis: Logistic regression modeled variables important for overall survival, transforming variables with multiple scales into dummy variables. Variables with a *p*-value < 0.05 were considered for the multivariate model.

Multivariate analysis: Univariate results informed a multivariate model, and the combinatorial analysis assessed interactions among variables. The models were evaluated based on the *p*-values, explained variance, sensitivity, and specificity. A five-variable model with optimal results was identified.

Nomogram classification: A nomogram, derived from the multivariate model, stratified recipients into two groups based on a cutoff point of 75 months of survival.

Donor factors and recipient prognosis: Logistic regression analyzed the donor variable associations with recipient prognosis based on the survival classification. No significant variables were found in the donor group.

Propensity score matching: Used to minimize selection bias, the model matched recipients based on parameters such as the sex, age, BMI, MELD, redo-LT, HCC, fulminant hepatitis indication, combined transplant, and vascular cause of the donor’s death, with a matching tolerance of 0.2 and a ratio of 1:1, as shown in Table 1.

Statistical analyses: These were conducted using R software version 4.3.1, with the tidyverse package suite for data manipulation, survival and survminer packages for survival analysis, and the caret package for machine learning techniques and predictive modeling.

## 3. Results

### 3.1. Characteristics of the Whole Donor Cohort (before PSM)

During the study period, the rate of donors over 85 y/o among all LTs was 3.2% and steadily increased from 1.2% in 2005 to 6.1% in 2022. In our study cohort, the eldest donor was 93 years old, the average was 87 (SD ± 2), and the median was 87 (IQR 85, 88). Between 2005 and 2023, 596 grafts from donors aged over 85 were proposed, while only 76 (12.7%) were accepted. Notably, 18 (23%) of these accepted grafts were “hors tour”, signifying livers initially declined by other centers but later endorsed by our transplant center after careful weighing of the benefits against the risks. Among the elderly donors, none reported drug use, with only one (1.3%) reporting alcohol consumption. A total of 13 (17.1%) donors had diabetes, 25 (32.9%) had cardiac pathologies, 5 (6.6%) had underlying pulmonary diseases, and 3 (3.9%) had nephropathy. Additionally, 21 (27.6%) donors presented with hypertension, while only 2 (2.6%) had a previous oncological history with acceptable risk. Importantly, the study prioritized selecting elderly donors with minimal comorbidities to evaluate the specific outcomes associated with elderly donor liver grafts. In the case of older donors, every effort was made to keep the CIT as short as possible. The median length of the CIT was 442 (IQR 376, 521).

#### 3.1.1. Elderly Donor Cohort (n = 76)

All recipients had cirrhosis, including hepatitis B virus infection-related cirrhosis in 12 (15.8%) patients, HCV infection-related cirrhosis in 8 (10.5%) patients, and alcohol-related cirrhosis in 34 (44.7%) patients. Thirty-six (47.4%) patients had HCC. The median MELD score at listing was 17 (IQR 10, 26). The median BMI of the donors was 26.1 (IQR 22; 29) kg/m2. Early allograft dysfunction and PNF occurred in 20 (26.3%) and 2 (2.6%) cases, respectively. Overall, biliary complications occurred in 12 (15.8%) patients, and hepatic artery complications in 7 (9.2%) patients. The rate of re-transplantation was 5.3% (n = 4), including early re-transplantation (PNF, n = 2) and late re-transplantation (chronic rejection, n = 1; Budd–Chiari syndrome = 1). For one PNF patient, transplantation was very difficult, with multiple transfusions (14 red blood cells [RBCs] transfused), portal thrombectomy, and parietal closure with VAC therapy due to hemodynamic instability and digestive bleeding. Graft biopsies of these two PNF patients did not reveal any worrisome features.

The median follow-up duration of the whole cohort was 39 (IQR 12; 63) months. One- and five-year patient survival rates were 90.2% and 63.3%, respectively. One- and five-year graft survival rates were 87.6% and 61%, respectively. The rate of re-transplantation was 5.3% (n = 4: 2 PNF, 1 chronic rejection, 1 Budd–Chiari), with all cases being re-transplanted during the first year.

#### 3.1.2. Young Donor Cohort (n = 349)

Among the recipients, 260 (74.5%) were diagnosed with cirrhosis, including 27 (8%) patients with hepatitis B virus infection-related cirrhosis, 66 (19%) patients with HCV infection-related cirrhosis, 102 (29%) patients with alcohol-related cirrhosis, 48 (14%) with cholestatic liver disorders, 38 (11%) with fulminant hepatitis, and 68 (19%) categorized under other indications (such as polycystic liver disease, autoimmune conditions, colorectal metastasis, etc.). Forty-four (12.6%) patients had HCC. The median MELD score at listing was 17 (IQR 12; 29). Primary non-function PNF occurred in 5 patients (1.4%). Overall, biliary complications occurred in 35 (10.3%) patients, and hepatic artery complications in 29 (8.3%) patients. The rate of re-transplantation was 2.8% (10 patients). Before PSM, the 5-year overall survival rates were 63% for the elderly group and 77% for the young group, respectively (log-rank test: *p* = 0.002; Wilcoxon test: *p* = 0.05).

### 3.2. Comparison of Recipient and Donor Characteristics after Propensity Score Matching (PSM)

An analysis of the recipient and donor characteristics revealed significant differences between the study groups. As shown in Table 2, in terms of recipient demographics, a notable gender discrepancy was observed, with the younger group comprising a higher proportion of male recipients compared to the older group (81.6% vs. 65.8%, *p* = 0.04). However, no significant variations were found in the age, BMI, MELD scores, or HCC prevalence between the two groups (*p* = 0.98 and *p* = 0.75, respectively). Similarly, the combined transplant rates and occurrences of fulminant hepatitis were comparable across groups. Conversely, the donor characteristics exhibited distinct disparities. Notably, a significantly lower percentage of male donors was observed in the older group compared to the younger group (25% vs. 72.4%, *p* < 0.001). The causes of donor death also varied significantly, with anoxia-related deaths being more prevalent in the older group (13.2%) and trauma-related deaths being more common in the younger group (50%) (*p* = 0.004 and *p* < 0.001, respectively). Moreover, vascular-related deaths were significantly higher in the older group (76.6%) compared to the younger group (15.8%) (*p* < 0.001). While the graft characteristics showed no significant differences in the steatosis content (*p* = 1), the younger group exhibited a higher graft-to-body-weight ratio. The CIT did not differ significantly between the groups. Notably, the D-MELD scores were significantly higher in the older group, with a greater proportion exceeding 1600 (*p* < 0.001).

### 3.3. Comparison of Perioperative Outcomes after LT in Groups after PSM

Comparing the post-LT outcomes between both groups revealed some discrepancies. In terms of the perioperative data, the older group experienced a higher median intraoperative RBC transfusion count (4 vs. 0 units, *p* = 0.065). However, the younger group had a longer median ICU length of stay (5 vs. 8 days, *p* < 0.001) and a shorter total length of stay (20 vs. 15 days, *p* < 0.001). The graft function outcomes demonstrated a trend toward higher rates of EAD in the older group (26.3% vs. 14.4%, *p* = 0.07), while the rates of PNF and rejection did not show significant differences. Regarding complications, there were no significant differences in the occurrence of ≥Grade III Clavien–Dindo complications, while the Comprehensive Complication Index (CCI) showed a trend toward higher values in the older group (mean of 22.9 vs. 16, *p* = 0.054). Biliary and vascular complications did not significantly differ between the groups. The perioperative outcomes after LT are summarized in Table 3.

The Kaplan–Meier estimation, showcased in Figure 1, illustrates the patient survival trends across donor age categories. Noteworthy distinctions emerged, as the one-year, three-year, and five-year patient survival rates reached 90.2%, 74.7%, and 63.3%, respectively, for the older donor group, and 85.2%, 80.8%, and 73.3% for the younger donor group (*p* = 0.1). The case control group manifested one-year, three-year, and five-year graft survival rates of 87.68%, 70.4%, and 61%, whereas the group control exhibited rates of 82.6%, 78.2%, and 72.7%, respectively (*p* = 0.1). These results underscore the remarkable outcomes observed at the 24-month mark with the use of grafts from donors aged ≥85 y/o, which were even lower compared to those from younger donors.

### 3.4. Intraoperative Features Influencing Outcomes after LT with Elderly Donor Liver Grafts

The operative variables in liver transplants using grafts from elderly donors were systematically evaluated. The parameters under scrutiny included the operative duration, transfusion needs, utilization of extracorporeal circulation, transjugular intrahepatic portosystemic shunt (TIPS), portal vein thrombosis, and ascites caval anastomosis techniques, arterial and portal anastomoses, biliodigestive reconstructions, and portal thrombosis occurrences. Notably, the only statistically significant influence on transplantation success was the number of transfusions required.

### 3.5. Outcomes after LT in Both Groups Compared with First-Year Benchmark Cutoffs

We compared recipients aged over 85 years with established benchmarks from the literature (based on low MELD recipients and low-risk LT data) [26] on low MELD recipients and low-risk LT data) [26] and an internal control group of donors under 40 years. The comprehensive comparison uncovered nuanced differences, particularly prolonged operative durations, increased transfusions, and extended ICU and hospital stays, along with elevated rates of re-transplantation and 1-year mortality in the study group. The comparison between the principal outcomes with the benchmark cutoff are shown in Table 4. In the elderly donor group, a Clavien >Grade III, biliary complications, and CCI score aligned with the established benchmarks; concurrently, the hospitalization days, intraoperative transfusions, re-transplants, and one-year mortality approached the upper limits proposed.

### 3.6. Preoperative Predictors of Graft and Patient Survivals

A univariate analysis revealed statistically significant associations between the 10-year survival and several pre-LT factors, including the AFP level (HR 1.008 [1–1.016], *p* = 0.046), waiting time on the transplant list (HR 1.002 [1.001–1.004], *p* = 0.006), recipient size (HR 0.946 [0.901–0.993], *p* = 0.026) and weight (HR 0.967 [0.939–0.996], *p* = 0.024), and tumor size (HR 1.022 [1.003–1.041], *p* = 0.023). The resulting nomogram, illustrating the predictive power for 10-year survival in cases where the donor is ≥85 y/o, is presented in Figure 2. If the nomogram yields less than 102 points, there is a hopeful outlook for a 90% five-year survival rate, as opposed to 25% for those with over 102 points, with a statistically significant difference (*p* < 0.0001) observed between the two survival curves. This underscores the potential prognostic value of the nomogram in predicting five-year survival outcomes, highlighting a substantial contrast in survival probabilities based on the calculated points.

## 4. Discussion

### 4.1. Statement of Principal Findings

The study contributes to the ongoing discourse by retrospectively analyzing liver transplants from donors aged over 85, focusing on a less-explored donor group. Our aim was to comprehensively understand the outcomes compared to a younger donor control group. Among recipients, cirrhosis was prevalent, with 47% having HCC. Biliary and hepatic artery complications occurred in 15.8% and 9.2% of cases. The re-transplantation rate was 5.3%, with one- and five-year patient survival rates of 90.2% and 63.3%. The graft function outcomes showed higher EAD rates in the older group. Comparisons with “ideal” LT benchmarks revealed nuances, and the Kaplan–Meier analysis indicated significant differences in patient survival before PSM (*p* = 0.002), which lessened after PSM (*p* = 0.1). A nomogram predicting the 10-year survival in donors ≥ 85 demonstrated potential prognostic value (*p* < 0.0001) based on pre-LT data.

### 4.2. Strengths and Weaknesses of the Study

A notable weakness of the study lies in its retrospective and single-center nature, resulting in a limited number of cases due to highly selective criteria for inclusion. However, it is worth noting that the study includes an almost unreported number of cases within this age range. To address potential confounding issues, a PSM approach was employed, which aimed to minimize the differences between the two groups and establish donor age as the likely causal factor influencing survival outcomes. It is important to acknowledge that despite using gender as an adjustment variable in the PSM process, we recognize the persistent difference in the recipient sex between the two groups. Due to the limited sample size and initial disparities, achieving a perfect balance proved challenging. However, since both groups predominantly consist of males, we believe this discrepancy does not significantly bias our comparison. The single-center nature of the study may limit its external validity, and residual confounding variables could persist despite PSM efforts. By comparing elderly donors with significantly younger individuals and avoiding reliance solely on standard benchmark cutoffs, the study ensures a nuanced analysis within the same time frame and consistent perioperative management protocols. Rigorous propensity score matching further emphasizes age as the predominant factor influencing patient outcomes.

### 4.3. Interpretation with Reference to Other Studies

The observed rise in the rate of donors aged over 85 years, escalating from 1.18% in 2005 to 6.11% in 2022, is indicative of a dynamic shift in the donor demographic landscape, necessitating a critical re-evaluation of transplantation strategies to align with the evolving trends in donor demographics and organ scarcity. Diverging from the prevailing literature, the present study highlights significant differences in the outcomes between younger and older donors, with an appreciable preference for the latter. These findings challenge the liberal use of grafts from older donor, highlighting the pivotal role of meticulous donor–recipient matching, particularly when considering elderly donors [27]. The ongoing global discourse surrounding the expansion of the age criterion for eligible liver donors has motivated varying degrees of enthusiasm and caution. Maestro O et al.’s study on nonagenarian donors provides insights (n = 6), yet our research suggests outcomes may significantly differ from younger counterparts, aligning with Haugen et al.’s findings [15].

The current study reinforces the importance of selection criteria for older donors and matching with the most correct recipients, fortifying the prudential attitude found in the literature, including studies by Pratschke et al. and Ghinolfi D et al., who advocate for specific combinations for optimal outcomes and propose the unrestricted evaluation of elderly donor grafts with active communication to the public [8,9]. However, unlike Pratschke et al. (Eurotransplant database), we did not highlight the same high-risk combinations, particularly old donors for HCV, high-MELD recipients, long CITs, or urgent LTs. We can justify this discrepancy by the low sample size for each of these situations, which is explained by our internal policy and logistical efforts to avoid these improper associations.

In the context of our surgical research, PSM was employed to balance the key variables between the two cohorts. The matching was based on crucial parameters, including the gender, age, BMI, MELD score, re-transplantation, HCC, fulminant hepatitis, combined transplantation, and vascular cause of death, as delineated in Table 1, as the main differences between the two groups.

It is evident that recipients of grafts from older donors exhibit a higher mortality rate. Among the 76 patients analyzed, 25 (33%) deceased to various causes. It is noteworthy that a significant portion, accounting for 18 patients (23%), experienced mortality due to reasons not directly related to the transplant procedure. These non-transplant-related factors encompassed events such as cardiovascular incidents and other unrelated causes. Additionally, 5 patients (7%) faced mortality due to tumor progression, while 2 patients (3%) died due to liver failure that resulted from a recurrence of their underlying liver pathology and not from complications related to the transplant. This means they experienced a cessation of liver function unrelated to either EAD or PNF.

These findings underscore the complex interplay of factors influencing post-transplant outcomes in recipients of grafts from elderly donors, with a need for further investigation into strategies to mitigate these risks and enhance patient survival, suggesting that matching according to the proposed nomogram would allow for excellent patient survival.

The present analysis reveals that one-, three-, and five-year graft and patient survival rates for older donors are suboptimal compared to those of their younger counterparts. While instances of EAD and PNF fall within acceptable limits, and although the literature highlights substantial rates of biliary and hepatic artery complications in the older donor cohort, our analysis demonstrated that these variances did not achieve statistical significance.

The introduction of a nomogram, a visual representation of prognostic factors influencing the 10-year survival after LT using a graft from a donor aged ≥85 years, adds a valuable dimension to our study. Surprisingly, the MELD score did not emerge as a significant predictor among these factors. This unexpected finding can be attributed to a meticulous pre-transplant selection process aimed at allocating older grafts to patients with less severe MELD scores (almost 50% of HCC). This insight is substantiated by data from the study period spanning 2005 to 2023, where 596 grafts aged over 85 were offered, but only 76 (12.7%) were accepted, and among these, 18 (23%) were categorized as “hors tour”. This remarkable correlation with the studies conducted by Pratschke et al. and Ghinolfi D et al. underscores the significance of donor age in LT. Even though age was modeled non-linearly, a clear linear trend was discerned, implying the absence of a distinct age cutoff. In clinical scenarios, it is advisable to approach combinations of older donor ages with factors like specific types of transplantation and hepatitis C with caution. While older grafts present potential benefits, such as reducing wait-list mortality, the idea of strict age cutoffs for brain-dead donors is being reconsidered [8,9]. This interesting finding can be linked to a meticulous pre-transplant selection process, mirroring the strategies employed in the aforementioned studies. Specifically, the careful allocation of older grafts to patients with low MELD scores played a pivotal role in the observed outcomes. The nomogram incorporates tumor features (AFP level and tumor size) and recipient morphological characteristics (size and weight), as well as the waiting time on the list, providing a comprehensive tool for predicting the 10-year survival in this specific LT setting. Here again, other criteria might have been expected, but these are the result of a largely pre-selected cohort.

In contrast to the existing literature, particularly the systematic review conducted by Domagala et al. [10], our current study reveals a different trend. Fewer biliary and vascular complications were observed with older grafts, and notably, this difference was not observed when compared to the younger group in our study. This unexpected outcome can be attributed to the meticulous selection process implemented during both the pre-transplant and allocation phase. Our findings align with Ghinolfi’s results on vascular complications in older donors [9] but differ from observations on biliary issues in younger recipients as reported by Jiménez [28]. Nardo et al. indicated no significant differences between older and younger groups in terms of acute rejection episodes requiring treatment and nonischemic biliary stenosis. Barbier’s study did not reveal significant disparities in EAD, PNF, hepatic artery, or biliary complications across age groups. Bertuzzo’s findings suggested that outcomes of LT with donors both ≥70 and <70 years are comparable with appropriate management. Additionally, fewer biliary complications were observed in the elderly according to Gajate et al.’s study, which contrasts with some other findings mentioned [13,14,29,30,31].

In this context, a comprehensive assessment of the intraoperative variables was conducted, mirroring the approach outlined by our group in Kitano’s research referring to unselected patients, “Subjective Difficulty Scale in Liver Transplantation” [32]. The current study’s findings indicates that in the meticulously chosen demographic for liver transplants with grafts from elderly donors, a transfusion requirement was the only intraoperative factor to be a statistically significant risk for survival. Notably, the features emphasized in Kitano’s paper, including annular segment one, late re-transplantation, and others, failed to demonstrate substantial correlations with transplant outcomes in this analysis. This distinction underscores the essential function of pre-transplant selection, extending beyond recipient characteristics to encompass the anticipation of technical challenges intrinsic to the procedure.

Despite the pre-selection process aiming to create a cohort with ostensibly favorable characteristics, both demographic and technical, the persistence of transfusion as a significant risk factor underscores its importance. Therefore, caution should be exercised, particularly in refraining from employing older grafts in challenging transplant scenarios, where high transfusion needs are anticipated (reflecting a higher risk of a longer CIT, hemodynamic instability, extra-corporeal circulation, etc., probably impacting function recovery of older grafts). Importantly, patients with HCC may present as optimal candidates in these situations, aligning with their potentially favorable outcomes amidst these specific challenges.

A basic message of the present study is the need for a more measured and cautious approach, emphasizing the potential value of HCC as an indication for LT. This underscores the imperative need for a rigorous donor evaluation process and a graft allocation policy that prioritizes recipients with a stable baseline functional status, particularly those dealing with HCC. This stance resonates with a broader body of the literature, consistently emphasizing the critical importance of judicious recipient–donor matching in optimizing outcomes, especially when utilizing grafts from older donors.

## 5. Conclusions

In conclusion, the present study underscores the significance of precise donor–recipient matching and prudent criterion selection, suggesting that HCC could potentially serve as a compelling indicator for the use of elderly donor grafts. On this condition, the results obtained after using a graft from donors >85 y.o. are quite acceptable in the context of the shortage. Despite the importance of addressing organ scarcity, it is essential to comprehend the complex dynamics of donor characteristics, rigorous selection methodologies, and advanced prognostic techniques. Such insights empower informed decision-making strategies within the evolving realm of liver transplantation.

## Figures and Tables

**Figure 1 cancers-16-01803-f001:**
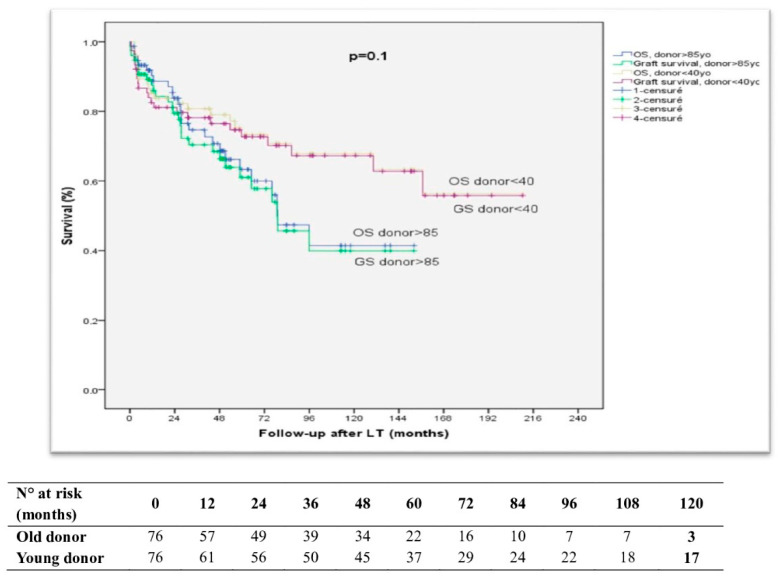
Kaplan–Meier curves depicting the influence of the donor age on outcomes in liver transplantation recipients (after PSM).

**Figure 2 cancers-16-01803-f002:**
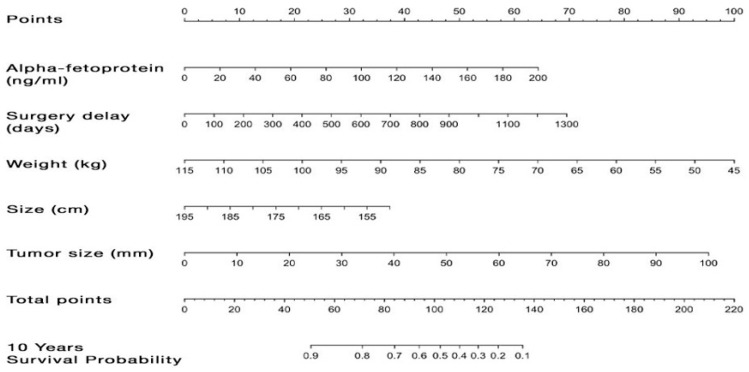
Nomogram illustrating donor–recipient matching optimization based on recipient data. The nomogram illustrates the predictive power for 10-year survival in cases where the donor is ≥85 years old. The variables used to calculate survival include the following: alpha-fetoprotein (AFP) level, waiting time on the transplant list, recipient size and weight, and tumor size. The dashed line on the nomogram correspond to the scores assigned to each variable based on their value. The sum of the total scores corresponds to the estimate of the probability of 10-year survival for the patient after liver transplantation.

**Table 1 cancers-16-01803-t001:** Demographical and clinical patient characteristics before and after propensity score matching (PSM).

	Before PSM			After PSM		
	Control Study (Donors ≤ 40 yo) n = 349	Study Group (Donors ≥ 85 yo) n = 76	*p*-Value	Control Study (Donors ≤ 40 yo) n = 76	Study Group (Donors ≥ 85 yo) n = 76	*p*-Value
Sex M (n, %)	212 (60.7%)	50 (65.8%)	0.44	62 (81.6%)	50 (65.8%)	0.04
Age (years, median, IQR)	44 [31, 59]	59 [54, 66]	<0.001	61 [56, 66]	59 [54, 66]	0.49
BMI (kg/m^2^, median, IQR)	24,6 [21, 27]	26.1 [22, 29]	0.02	25.6 [22, 29]	26.1 [22, 29]	0.55
MELD (median, IQR)	17 [12, 29]	17 [10, 26]	0.91	14.3 [10, 25]	17 [10, 26]	0.98
ReLT (n, %)	83 (23.8%)	4 (5.3%)	0.001	5 (6.6%)	4 (5.3%)	1
HCC (n, %)	44 (12.6%)	36 (47.4%)	<0.001	39 (51.3%)	36 (47.4%)	0.75
Fulminant hepatitis (n, %)	32 (9.2%)	6 (7.9%)	0.53	32 (24.3%)	6 (7.9%)	0.53
Combined transplant (n, %)	37 (10.6%)	0 (0%)	0.001	0 (0%)	0 (0%)	NA
Vascular cause of death (n, %)	65 (18.6%)	61 (80.3%)	<0.001	10 (13.2%)	61 (80.3%)	<0.001

Abbreviations: BMI, body mass index; HCC, hepatocellular carcinoma; MELD, Model for End-Stage Liver Disease; ReLT, redo transplant.

**Table 2 cancers-16-01803-t002:** Recipient and donor characteristics in the older group and in the younger group after PSM.

		Younger Group (n = 76)	Older Group (n = 76)	*p*-Value
Recipient demographics	Sex M (n, %)	62 (81.6%)	50 (65.8%)	0.04
	Age, y (median, IQR)	61 [56, 66]	59 [54, 66]	0.49
	BMI (median, IQR)	25.6 [22, 29]	26.1 [22, 29]	0.55
Recipient underlying hepatopathy	MELD (median, IQR)	16 [11, 27]	17 [10, 26]	98
	HCC (n, %)	39 (51.3%)	36 (47.4%)	0.75
Type of LT	ReLT (n, %)	5 (6.6%)	4 (5.3%)	1
	Fulminant hepatitis (n, %)	32 (24.3%)	6 (7.9%)	0.53
Donor characteristics	Sex M (n, %)	55 (72.4%)	19 (25%)	<0.001
	BMI (median, IQR)	23.5 [21, 26)	23.2 [21, 26]	0.62
Donor cause of death	Anoxia	26 (34.2%)	10 (13.2%)	0.004
	Trauma	38 (50%)	15 (19.7%)	<0.001
	Vascular	12 (15.8%)	50 (76.6%)	<0.001
Graft characteristics	Steatosis * ≤ 30% (n, %)	72 (95%)	64 (94%) **	1
	GRWR (mean ± IQR)	1.94 [1.6, 2.8]	1.4 [1.2, 1.7]	<0.001
	CIT, min (mean ± IQR)	442 [376, 521]	416 [371, 483]	0.40
D-MELD score	D-MELD score value	458 [300, 764]	1505 [840, 2306]	<0.001
	D-MELD score > 1600 (n, %)	1 (1.3%)	36 (47.4%)	<0.001

Abbreviations: BMI, body mass index; CIT, cold ischemia time; GRWR, graft-to-recipient-weight ratio; HCC, hepatocellular carcinoma; MELD, Model for End-Stage Liver Disease; ReLT, re-transplant. * Graft steatosis was assessed on the biopsy performed during the procurement (definitive analysis). ** The percentage of steatosis in elderly donors was conducted on a total of 68 patients due to a lack of data.

**Table 3 cancers-16-01803-t003:** Perioperative outcomes after LT in the older group and in the younger group (after propensity matching).

Outcomes		Younger Group (n = 76)	Older Group (n = 76)	*p*-Value
Perioperative data	Intraoperative RBC transfusions (median, IQR)	0 [0, 7]	4 [0, 6]	0.065
	ICU length of stay (median, IQR)	8 [5, 14]	5 [4, 9]	<0.001
	Total length of stay (median, IQR)	15 [9, 21]	20 [15, 32]	<0.001
Graft function	EAD (n, %)	11 (14.4)	20 (26.3)	0.07
	PNF (n, %)	0 (0)	2 (2.6)	0.50
	Rejection (n, %)	2 (2.62)	6 (7.9)	0.17
Complications	≥Grade III Clavien–Dindo score (n, %)	22 (28.9)	21 (27.6)	1
	CCI (mean, ±SD)	16 (23.1)	22.9 (22.3)	0.054
	Biliary complications (n, %)	7 (9.21)	7 (9.21)	0.33
	Vascular complications (n, %)	14 (18.4)	8 (10.5)	0.10
Patient survival	1 y survival rates (95% CI)	85.2%	90.2%	0.1
	3 y survival rates (95% CI)	80.8%	74.7%	
	5 y survival rates (95% CI)	73.3%	63.3%	
Graft survival	1 y survival rates (95% CI)	82.6%	87.6%	
	3 y survival rates (95% CI)	78.2%	70.4%	
	5 y survival rates (95% CI)	72.7%	61%	

Abbreviations: CCI, Comprehensive Complication Index; EAD, early allograft dysfunction; ICU, intensive care unit; PNF, Comprehensive Complication Index; RBC Red Blood Cell.

**Table 4 cancers-16-01803-t004:** First-year outcomes after liver transplantation in two groups compared with first-year benchmark cutoffs.

	Case (>85 yo)	Benchmark Cutoffs (at 12 Months)	Control (<40 yo)
MELD	17 [10, 26]	12 [9, 16]	14 [10, 25]
OP duration (h), (median, IQR)	7.1 (6.4, 8.1]	≤6	7 [6, 7.9]
Intraoperative RBC transfusions (median, IQR)	4 (0, 6)	≤3	0 (0, 7)
ICU stays (d), (median, IQR)	5 (4, 9)	≤4	8 (5, 14)
Hospital stays (d), (median, IQR)	20 (15, 32)	≤18	15 (9, 21)
≥Grade III Clavien–Dindo score (%)	27.6	≤59	28.9
Biliary complications, (%)	9.2	≤28	9.21
Re-transplantations, (%)	5.3	≤4	2.8
CCI (mean, ±SD)	22.9 (22.3)	≤29.6	16 (23.1)
1-year mortality, (%)	10.5	≤9	14.4

Abbreviations: CCI, Comprehensive Complication Index; d, days; h, hours; ICU, intensive care unit; MELD, Model for End-Stage Liver Disease; OP, operative; RBC, red blood cell.

## Data Availability

The data that support the findings of this study are available from Paul-Brousse/Bicêtre hospital upon reasonable request. Restrictions apply to the availability of these data, which were used under license for this study.

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
