# Peer review of "Liver Transplantation from Elderly Donors (≥85 Years Old)"

_cancers, 2024, doi:10.3390/cancers16101803_

Round 1
Reviewer 1 Report
Comments and Suggestions for Authors
The authors explored the viability of liver grafts from very old donors (>85 years old) in comparison to younger donors under 40 years of age. The theme is highly intriguing and holds significant importance in expanding the donor pool. However, there are several concerns that warrant attention:
Major Revisions
- Propensity Score Matching Variables
The variables utilized in the Propensity Score Matching (PSM) process require clarification. Even after matching, a significant difference in recipient sex persisted, which could potentially influence the prognosis after liver transplantation. It is recommended to include recipient sex as a variable in the PSM analysis to account for this potential confounding factor. - Kaplan-Meier Evaluation and Interpretation
The results of the Kaplan-Meier evaluation appear to deviate from the presented interpretation. It is advisable to provide the results from both the log-rank and Wilcoxon analyses for a more comprehensive understanding. The Kaplan-Meier analysis suggests equivalent prognosis in the first 24-36 months, with the prognosis from older donors becoming inferior thereafter. This finding aligns with theoretical expectations and does not necessarily imply that these grafts are too risky to accept. These results offer valuable insights for clinical practice. Based on a precise interpretation of the results, the authors should clarify which recipient group is likely to benefit from transplants involving older donor grafts. - Preoperative Predictors of Graft and Patient Survival
The methodology and underlying logistics of the analysis demonstrating preoperative predictors of graft and patient survival are challenging to comprehend. It is recommended to exhibit the results of the univariate analysis to enhance clarity and understanding. Alternatively, if the analysis proves too complex to convey effectively, it may be advisable to consider removing this analysis from the manuscript.
Author Response
- Propensity Score Matching Variables
The variables utilized in the Propensity Score Matching (PSM) process require clarification. Even after matching, a significant difference in recipient sex persisted, which could potentially influence the prognosis after liver transplantation. It is recommended to include recipient sex as a variable in the PSM analysis to account for this potential confounding factor.
Your feedback is greatly appreciated. Despite using gender as an adjustment variable in the Propensity Score Matching (PSM) process, we acknowledge the persistent difference in recipient sex between the two groups. Given the limited cohort size and initial disparities, achieving perfect balance proved challenging. However, as both groups predominantly consist of men, we believe this discrepancy does not significantly bias our comparison. We appreciate your attention to detail and welcome any further suggestions. In the article discussion, we will underline this limitation that you raised.
- Kaplan-Meier Evaluation and Interpretation
The results of the Kaplan-Meier evaluation appear to deviate from the presented interpretation. It is advisable to provide the results from both the log-rank and Wilcoxon analyses for a more comprehensive understanding. The Kaplan-Meier analysis suggests equivalent prognosis in the first 24-36 months, with the prognosis from older donors becoming inferior thereafter. This finding aligns with theoretical expectations and does not necessarily imply that these grafts are too risky to accept. These results offer valuable insights for clinical practice. Based on a precise interpretation of the results, the authors should clarify which recipient group is likely to benefit from transplants involving older donor grafts.
Thank you for your careful analysis. We performed the Gehan-Breslow-Wilcoxon test as suggested, both before and after Propensity Score Matching (PSM).
- Before PSM, the p-value was 0.05, consistent with our previous log-rank test (p=0.002).
- After PSM, the p-value was 0.75, aligning with the log-rank result (p=0.1).
These findings validate our previous results, indicating poorer long-term outcomes with aged grafts. We will emphasize this in the discussion, along with clarifying which recipient group benefits most from older donor grafts.
- Preoperative Predictors of Graft and Patient Survival
The methodology and underlying logistics of the analysis demonstrating preoperative predictors of graft and patient survival are challenging to comprehend. It is recommended to exhibit the results of the univariate analysis to enhance clarity and understanding. Alternatively, if the analysis proves too complex to convey effectively, it may be advisable to consider removing this analysis from the manuscript.
We opted not to include a specific table and instead incorporated the p-values and hazard ratios (HR) directly into the text. This decision was made to enhance readability and streamline the presentation of results. Additionally, we chose not to include an additional figure as this analysis serves as a secondary/intermediate step in explaining the methodology behind the nomogram creation. Our focus remains on clarity and conciseness in conveying the intricacies of the analysis.
Reviewer 2 Report
Comments and Suggestions for Authors
While this is not completely novel as noted in the references cited, this is one of the larger studies looking at older donors for liver transplant and the authors have divided the donors into two discrete groups. Performing PSM is a good strategy in this retrospective study and helps adjust for the extended time period of the study and variability/bias that may occur.
Author Response
While this is not completely novel as noted in the references cited, this is one of the larger studies looking at older donors for liver transplant and the authors have divided the donors into two discrete groups. Performing PSM is a good strategy in this retrospective study and helps adjust for the extended time period of the study and variability/bias that may occur.
Thank you for your review and positive support. We acknowledge your observation regarding the existing literature and appreciate your recognition of the significance of our study in examining older donors for liver transplant. We agree that employing Propensity Score Matching (PSM) in our retrospective study is a valuable strategy to address potential variability and bias. Your feedback reinforces the importance of our research in advancing understanding within the field of liver transplantation.